# Improved Variational Inference
# with Inverse Autoregressive Flow

**Diederik P. Kingma**
dpkingma@openai.com

**Tim Salimans**
tim@openai.com

**Rafal Jozefowicz**
rafal@openai.com

**Xi Chen**
peter@openai.com

**Ilya Sutskever**
ilya@openai.com

**Max Welling**[*]
M.Welling@uva.nl

## Abstract

The framework of normalizing flows provides a general strategy for flexible variational inference of posteriors over latent variables. We propose a new type of normalizing flow, inverse autoregressive flow (IAF), that, in contrast to earlier published flows, scales well to high-dimensional latent spaces. The proposed flow consists of a chain of invertible transformations, where each transformation is based on an autoregressive neural network. In experiments, we show that IAF significantly improves upon diagonal Gaussian approximate posteriors. In addition, we demonstrate that a novel type of variational autoencoder, coupled with IAF, is competitive with neural autoregressive models in terms of attained log-likelihood on natural images, while allowing significantly faster synthesis.

## 1 Introduction

Stochastic variational inference (Blei et al., 2012; Hoffman et al., 2013) is a method for scalable posterior inference with large datasets using stochastic gradient ascent. It can be made especially efficient for continuous latent variables through latent-variable reparameterization and inference networks, amortizing the cost, resulting in a highly scalable learning procedure (Kingma and Welling, 2013; Rezende et al., 2014; Salimans et al., 2014). When using neural networks for both the inference network and generative model, this results in class of models called variational auto-encoders (Kingma and Welling, 2013) (VAEs). A general strategy for building flexible inference networks, is the framework of *normalizing flows* (Rezende and Mohamed, 2015). In this paper we propose a new type of flow, *inverse autoregressive flow* (IAF), which scales well to high-dimensional latent space.

At the core of our proposed method lie Gaussian autoregressive functions that are normally used for density estimation: functions that take as input a variable with some specified ordering such as multidimensional tensors, and output a mean and standard deviation for each element of the input variable conditioned on the previous elements. Examples of such functions are autoregressive neural density estimators such as RNNs, MADE (Germain et al., 2015), PixelCNN (van den Oord et al., 2016b) or WaveNet (van den Oord et al., 2016a) models. We show that such functions can often be turned into invertible nonlinear transformations of the input, with a simple Jacobian determinant. Since the transformation is flexible and the determinant known, it can be used as a normalizing flow, transforming a tensor with relatively simple known density, into a new tensor with more complicated density that is still cheaply computable. In contrast with most previous work on

---

[*]University of Amsterdam, University of California Irvine, and the Canadian Institute for Advanced Research (CIFAR).

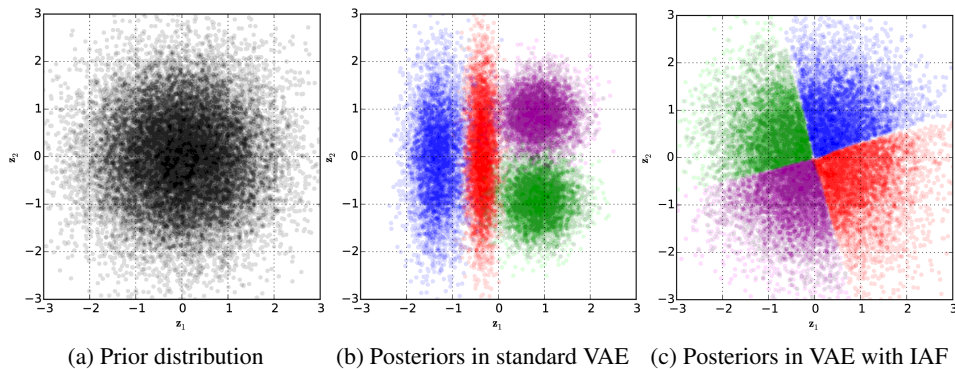

(a) Prior distribution     (b) Posteriors in standard VAE    (c) Posteriors in VAE with IAF

Figure 1: Best viewed in color. We fitted a variational auto-encoder (VAE) with a spherical Gaussian prior, and with factorized Gaussian posteriors **(b)** or inverse autoregressive flow (IAF) posteriors **(c)** to a toy dataset with four datapoints. Each colored cluster corresponds to the posterior distribution of one datapoint. IAF greatly improves the flexibility of the posterior distributions, and allows for a much better fit between the posteriors and the prior.

improving inference models including previously used normalizing flows, this transformation is well suited to high-dimensional tensor variables, such as spatio-temporally organized variables.

We demonstrate this method by improving inference networks of deep variational auto-encoders. In particular, we train deep variational auto-encoders with latent variables at multiple levels of the hierarchy, where each stochastic variable is a three-dimensional tensor (a stack of featuremaps), and demonstrate improved performance.

## 2   Variational Inference and Learning

Let $\mathbf{x}$ be a (set of) observed variable(s), $\mathbf{z}$ a (set of) latent variable(s) and let $p(\mathbf{x}, \mathbf{z})$ be the parametric model of their joint distribution, called the *generative model* defined over the variables. Given a dataset $\mathbf{X} = \{\mathbf{x}^1, ..., \mathbf{x}^N\}$ we typically wish to perform maximum marginal likelihood learning of its parameters, i.e. to maximize

$$\log p(\mathbf{X}) = \sum_{i=1}^{N} \log p(\mathbf{x}^{(i)}), \tag{1}$$

but in general this marginal likelihood is intractable to compute or differentiate directly for flexible generative models, e.g. when components of the generative model are parameterized by neural networks. A solution is to introduce $q(\mathbf{z}|\mathbf{x})$, a parametric *inference model* defined over the latent variables, and optimize the *variational lower bound* on the marginal log-likelihood of each observation $\mathbf{x}$:

$$\log p(\mathbf{x}) \geq \mathbb{E}_{q(\mathbf{z}|\mathbf{x})}\left[\log p(\mathbf{x}, \mathbf{z}) - \log q(\mathbf{z}|\mathbf{x})\right] = \mathcal{L}(\mathbf{x}; \boldsymbol{\theta}) \tag{2}$$

where $\boldsymbol{\theta}$ indicates the parameters of $p$ and $q$ models. Keeping in mind that Kullback-Leibler divergences $D_{KL}(.)$ are non-negative, it's clear that $\mathcal{L}(\mathbf{x}; \boldsymbol{\theta})$ is a lower bound on $\log p(\mathbf{x})$ since it can be written as follows ):

$$\mathcal{L}(\mathbf{x}; \boldsymbol{\theta}) = \log p(\mathbf{x}) - D_{KL}(q(\mathbf{z}|\mathbf{x})||p(\mathbf{z}|\mathbf{x})) \tag{3}$$

There are various ways to optimize the lower bound $\mathcal{L}(\mathbf{x}; \boldsymbol{\theta})$; for continuous $\mathbf{z}$ it can be done efficiently through a re-parameterization of $q(\mathbf{z}|\mathbf{x})$, see e.g. (Kingma and Welling, 2013; Rezende et al., 2014).

As can be seen from equation (3), maximizing $\mathcal{L}(\mathbf{x}; \boldsymbol{\theta})$ w.r.t. $\boldsymbol{\theta}$ will concurrently maximize $\log p(\mathbf{x})$ and minimize $D_{KL}(q(\mathbf{z}|\mathbf{x})||p(\mathbf{z}|\mathbf{x}))$. The closer $D_{KL}(q(\mathbf{z}|\mathbf{x})||p(\mathbf{z}|\mathbf{x}))$ is to 0, the closer $\mathcal{L}(\mathbf{x}; \boldsymbol{\theta})$ will be to $\log p(\mathbf{x})$, and the better an approximation our optimization objective $\mathcal{L}(\mathbf{x}; \boldsymbol{\theta})$ is to our true objective $\log p(\mathbf{x})$. Also, minimization of $D_{KL}(q(\mathbf{z}|\mathbf{x})||p(\mathbf{z}|\mathbf{x}))$ can be a goal in itself, if we're interested in using $q(\mathbf{z}|\mathbf{x})$ for inference after optimization. In any case, the divergence $D_{KL}(q(\mathbf{z}|\mathbf{x})||p(\mathbf{z}|\mathbf{x}))$ is a function of our parameters through both the inference model and the generative model, and increasing the flexibility of either is generally helpful towards our objective.

Note that in models with multiple latent variables, the inference model is typically factorized into partial inference models with some ordering; e.g. $q(\mathbf{z}_a, \mathbf{z}_b|\mathbf{x}) = q(\mathbf{z}_a|\mathbf{x})q(\mathbf{z}_b|\mathbf{z}_a, \mathbf{x})$. We'll write $q(\mathbf{z}|\mathbf{x}, \mathbf{c})$ to denote such partial inference models, conditioned on both the data $\mathbf{x}$ and a further context $\mathbf{c}$ which includes the previous latent variables according to the ordering.

## 2.1 Requirements for Computational Tractability

Requirements for the inference model, in order to be able to efficiently optimize the bound, are that it is (1) computationally efficient to compute and differentiate its probability density $q(\mathbf{z}|\mathbf{x})$, and (2) computationally efficient to sample from, since both these operations need to be performed for each datapoint in a minibatch at every iteration of optimization. If $\mathbf{z}$ is high-dimensional and we want to make efficient use of parallel computational resources like GPUs, then parallelizability of these operations across dimensions of $\mathbf{z}$ is a large factor towards efficiency. This requirement restrict the class of approximate posteriors $q(\mathbf{z}|\mathbf{x})$ that are practical to use. In practice this often leads to the use of diagonal posteriors, e.g. $q(\mathbf{z}|\mathbf{x}) \sim \mathcal{N}(\boldsymbol{\mu}(\mathbf{x}), \boldsymbol{\sigma}^2(\mathbf{x}))$, where $\boldsymbol{\mu}(\mathbf{x})$ and $\boldsymbol{\sigma}(\mathbf{x})$ are often nonlinear functions parameterized by neural networks. However, as explained above, we also need the density $q(\mathbf{z}|\mathbf{x})$ to be sufficiently flexible to match the true posterior $p(\mathbf{z}|\mathbf{x})$.

## 2.2 Normalizing Flow

*Normalizing Flow* (NF), introduced by (Rezende and Mohamed, 2015) in the context of stochastic gradient variational inference, is a powerful framework for building flexible posterior distributions through an iterative procedure. The general idea is to start off with an initial random variable with a relatively simple distribution with known (and computationally cheap) probability density function, and then apply a chain of invertible parameterized transformations $\mathbf{f}_t$, such that the last iterate $\mathbf{z}_T$ has a more flexible distribution[2]:

$$\mathbf{z}_0 \sim q(\mathbf{z}_0|\mathbf{x}), \quad \mathbf{z}_t = \mathbf{f}_t(\mathbf{z}_{t-1}, \mathbf{x}) \quad \forall t = 1...T \tag{4}$$

As long as the Jacobian determinant of each of the transformations $\mathbf{f}_t$ can be computed, we can still compute the probability density function of the last iterate:

$$\log q(\mathbf{z}_T|\mathbf{x}) = \log q(\mathbf{z}_0|\mathbf{x}) - \sum_{t=1}^{T} \log \det \left| \frac{d\mathbf{z}_t}{d\mathbf{z}_{t-1}} \right| \tag{5}$$

However, (Rezende and Mohamed, 2015) experiment with only a very limited family of such invertible transformation with known Jacobian determinant, namely:

$$\mathbf{f}_t(\mathbf{z}_{t-1}) = \mathbf{z}_{t-1} + \mathbf{u}h(\mathbf{w}^T\mathbf{z}_{t-1} + b) \tag{6}$$

where $\mathbf{u}$ and $\mathbf{w}$ are vectors, $\mathbf{w}^T$ is $\mathbf{w}$ transposed, $b$ is a scalar and $h(.)$ is a nonlinearity, such that $\mathbf{u}h(\mathbf{w}^T\mathbf{z}_{t-1} + b)$ can be interpreted as a MLP with a bottleneck hidden layer with a single unit. Since information goes through the single bottleneck, a long chain of transformations is required to capture high-dimensional dependencies.

## 3 Inverse Autoregressive Transformations

In order to find a type of normalizing flow that scales well to high-dimensional space, we consider Gaussian versions of autoregressive autoencoders such as MADE (Germain et al., 2015) and the PixelCNN (van den Oord et al., 2016b). Let $\mathbf{y}$ be a variable modeled by such a model, with some chosen ordering on its elements $\mathbf{y} = \{y_i\}_{i=1}^{D}$. We will use $[\boldsymbol{\mu}(\mathbf{y}), \boldsymbol{\sigma}(\mathbf{y})]$ to denote the function of the vector $\mathbf{y}$, to the vectors $\boldsymbol{\mu}$ and $\boldsymbol{\sigma}$. Due to the autoregressive structure, the Jacobian is lower triangular with zeros on the diagonal: $\partial[\boldsymbol{\mu}_i, \boldsymbol{\sigma}_i]/\partial \mathbf{y}_j = [0, 0]$ for $j \geq i$. The elements $[\mu_i(\mathbf{y}_{1:i-1}), \sigma_i(\mathbf{y}_{1:i-1})]$ are the predicted mean and standard deviation of the $i$-th element of $\mathbf{y}$, which are functions of only the previous elements in $\mathbf{y}$.

Sampling from such a model is a sequential transformation from a noise vector $\boldsymbol{\epsilon} \sim \mathcal{N}(0, \mathbf{I})$ to the corresponding vector $\mathbf{y}$: $y_0 = \mu_0 + \sigma_0 \odot \epsilon_0$, and for $i > 0$, $y_i = \mu_i(\mathbf{y}_{1:i-1}) + \sigma_i(\mathbf{y}_{1:i-1}) \cdot \epsilon_i$. The

**Algorithm 1:** Pseudo-code of an approximate posterior with Inverse Autoregressive Flow (IAF)

**Data**:
    $\mathbf{x}$: a datapoint, and optionally other conditioning information
    $\boldsymbol{\theta}$: neural network parameters
    $\texttt{EncoderNN}(\mathbf{x}; \boldsymbol{\theta})$: encoder neural network, with additional output $\mathbf{h}$
    $\texttt{AutoregressiveNN}[*](\mathbf{z}, \mathbf{h}; \boldsymbol{\theta})$: autoregressive neural networks, with additional input $\mathbf{h}$
    $\texttt{sum}(.)$: sum over vector elements
    $\texttt{sigmoid}(.)$: element-wise sigmoid function

**Result**:
    $\mathbf{z}$: a random sample from $q(\mathbf{z}|\mathbf{x})$, the approximate posterior distribution
    $l$: the scalar value of $\log q(\mathbf{z}|\mathbf{x})$, evaluated at sample '$\mathbf{z}$'

$[\boldsymbol{\mu}, \boldsymbol{\sigma}, \mathbf{h}] \leftarrow \texttt{EncoderNN}(\mathbf{x}; \boldsymbol{\theta})$
$\boldsymbol{\epsilon} \sim \mathcal{N}(0, I)$
$\mathbf{z} \leftarrow \boldsymbol{\sigma} \odot \boldsymbol{\epsilon} + \boldsymbol{\mu}$
$l \leftarrow -\texttt{sum}(\log \boldsymbol{\sigma} + \frac{1}{2}\boldsymbol{\epsilon}^2 + \frac{1}{2}\log(2\pi))$
**for** $t \leftarrow 1$ **to** $T$ **do**
    $[\mathbf{m}, \mathbf{s}] \leftarrow \texttt{AutoregressiveNN}[t](\mathbf{z}, \mathbf{h}; \boldsymbol{\theta})$
    $\boldsymbol{\sigma} \leftarrow \texttt{sigmoid}(\mathbf{s})$
    $\mathbf{z} \leftarrow \boldsymbol{\sigma} \odot \mathbf{z} + (1 - \boldsymbol{\sigma}) \odot \mathbf{m}$
    $l \leftarrow l - \texttt{sum}(\log \boldsymbol{\sigma})$
**end**

---

computation involved in this transformation is clearly proportional to the dimensionality $D$. Since variational inference requires sampling from the posterior, such models are not interesting for direct use in such applications. However, the inverse transformation is interesting for normalizing flows, as we will show. As long as we have $\sigma_i > 0$ for all $i$, the sampling transformation above is a one-to-one transformation, and can be inverted: $\epsilon_i = \frac{y_i - \mu_i(\mathbf{y}_{1:i-1})}{\sigma_i(\mathbf{y}_{1:i-1})}$.

We make two key observations, important for normalizing flows. The first is that this inverse transformation can be parallelized, since (in case of autoregressive autoencoders) computations of the individual elements $\epsilon_i$ do not depend on eachother. The vectorized transformation is:

$$\boldsymbol{\epsilon} = (\mathbf{y} - \boldsymbol{\mu}(\mathbf{y}))/\boldsymbol{\sigma}(\mathbf{y}) \tag{7}$$

where the subtraction and division are elementwise.

The second key observation, is that this inverse autoregressive operation has a simple Jacobian determinant. Note that due to the autoregressive structure, $\partial[\mu_i, \sigma_i]/\partial y_j = [0, 0]$ for $j \geq i$. As a result, the transformation has a lower triangular Jacobian ($\partial \epsilon_i/\partial y_j = 0$ for $j > i$), with a simple diagonal: $\partial \epsilon_i/\partial y_i = \sigma_i$. The determinant of a lower triangular matrix equals the product of the diagonal terms. As a result, the log-determinant of the Jacobian of the transformation is remarkably simple and straightforward to compute:

$$\log \det \left| \frac{d\boldsymbol{\epsilon}}{d\mathbf{y}} \right| = \sum_{i=1}^{D} -\log \sigma_i(\mathbf{y}) \tag{8}$$

The combination of model flexibility, parallelizability across dimensions, and simple log-determinant, make this transformation interesting for use as a normalizing flow over high-dimensional latent space.

## 4 Inverse Autoregressive Flow (IAF)

We propose a new type normalizing flow (eq. (5)), based on transformations that are equivalent to the inverse autoregressive transformation of eq. (7) up to reparameterization. See algorithm 1 for pseudo-code of an appproximate posterior with the proposed flow. We let an initial encoder neural network output $\boldsymbol{\mu}_0$ and $\boldsymbol{\sigma}_0$, in addition to an extra output $\mathbf{h}$, which serves as an additional input to each subsequent step in the flow. We draw a random sample $\boldsymbol{\epsilon} \sim \mathcal{N}(0, I)$, and initialize the chain with:

$$\mathbf{z}_0 = \boldsymbol{\mu}_0 + \boldsymbol{\sigma}_0 \odot \boldsymbol{\epsilon} \tag{9}$$

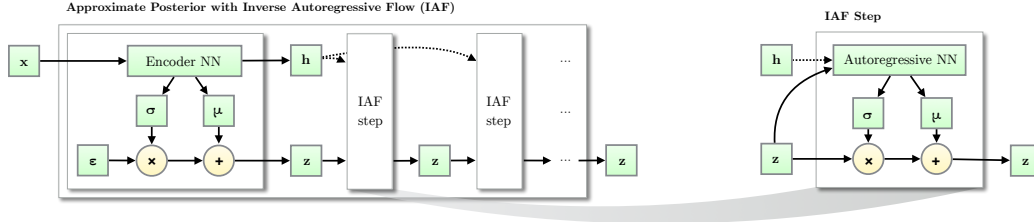

Figure 2: Like other normalizing flows, drawing samples from an approximate posterior with Inverse Autoregressive Flow (IAF) consists of an initial sample $\mathbf{z}$ drawn from a simple distribution, such as a Gaussian with diagonal covariance, followed by a chain of nonlinear invertible transformations of $\mathbf{z}$, each with a simple Jacobian determinants.

The flow consists of a chain of $T$ of the following transformations:

$$\mathbf{z}_t = \boldsymbol{\mu}_t + \boldsymbol{\sigma}_t \odot \mathbf{z}_{t-1} \tag{10}$$

where at the $t$-th step of the flow, we use a different autoregressive neural network with inputs $\mathbf{z}_{t-1}$ and $\mathbf{h}$, and outputs $\boldsymbol{\mu}_t$ and $\boldsymbol{\sigma}_t$. The neural network is structured to be autoregressive w.r.t. $\mathbf{z}_{t-1}$, such that for any choice of its parameters, the Jacobians $\frac{d\boldsymbol{\mu}_t}{d\mathbf{z}_{t-1}}$ and $\frac{d\boldsymbol{\sigma}_t}{d\mathbf{z}_{t-1}}$ are triangular with zeros on the diagonal. As a result, $\frac{d\mathbf{z}_t}{d\mathbf{z}_{t-1}}$ is triangular with $\boldsymbol{\sigma}_t$ on the diagonal, with determinant $\prod_{i=1}^{D} \sigma_{t,i}$. (Note that the Jacobian w.r.t. $\mathbf{h}$ does not have constraints.) Following eq. (5), the density under the final iterate is:

$$\log q(\mathbf{z}_T|\mathbf{x}) = -\sum_{i=1}^{D}\left(\tfrac{1}{2}\epsilon_i^2 + \tfrac{1}{2}\log(2\pi) + \sum_{t=0}^{T}\log\sigma_{t,i}\right) \tag{11}$$

The flexibility of the distribution of the final iterate $\mathbf{z}_T$, and its ability to closely fit to the true posterior, increases with the expressivity of the autoregressive models and the depth of the chain. See figure 2 for an illustration.

A numerically stable version, inspired by the LSTM-type update, is where we let the autoregressive network output $[\mathbf{m}_t, \mathbf{s}_t]$, two unconstrained real-valued vectors:

$$[\mathbf{m}_t, \mathbf{s}_t] \leftarrow \texttt{AutoregressiveNN}[t](\mathbf{z}_t, \mathbf{h}; \boldsymbol{\theta}) \tag{12}$$

and compute $\mathbf{z}_t$ as:

$$\boldsymbol{\sigma}_t = \mathrm{sigmoid}(\mathbf{s}_t) \tag{13}$$
$$\mathbf{z}_t = \boldsymbol{\sigma}_t \odot \mathbf{z}_{t-1} + (1 - \boldsymbol{\sigma}_t) \odot \mathbf{m}_t \tag{14}$$

This version is shown in algorithm 1. Note that this is just a particular version of the update of eq. (10), so the simple computation of the final log-density of eq. (11) still applies.

We found it beneficial for results to parameterize or initialize the parameters of each `AutoregressiveNN`[$t$] such that its outputs $\mathbf{s}_t$ are, before optimization, sufficiently positive, such as close to +1 or +2. This leads to an initial behaviour that updates $\mathbf{z}$ only slightly with each step of IAF. Such a parameterization is known as a 'forget gate bias' in LSTMs, as investigated by Jozefowicz et al. (2015).

Perhaps the simplest special version of IAF is one with a simple step, and a linear autoregressive model. This transforms a Gaussian variable with diagonal covariance, to one with linear dependencies, i.e. a Gaussian distribution with full covariance. See appendix A for an explanation.

Autoregressive neural networks form a rich family of nonlinear transformations for IAF. For non-convolutional models, we use the family of masked autoregressive networks introduced in (Germain et al., 2015) for the autoregressive neural networks. For CIFAR-10 experiments, which benefits more from scaling to high dimensional latent space, we use the family of convolutional autoregressive autoencoders introduced by (van den Oord et al., 2016b,c).

We found that results improved when reversing the ordering of the variables after each step in the IAF chain. This is a volume-preserving transformation, so the simple form of eq. (11) remains unchanged.

# 5 Related work

Inverse autoregressive flow (IAF) is a member of the family of normalizing flows, first discussed in (Rezende and Mohamed, 2015) in the context of stochastic variational inference. In (Rezende and Mohamed, 2015) two specific types of flows are introduced: planar flows and radial flows. These flows are shown to be effective to problems relatively low-dimensional latent space (at most a few hundred dimensions). It is not clear, however, how to scale such flows to much higher-dimensional latent spaces, such as latent spaces of generative models of /larger images, and how planar and radial flows can leverage the topology of latent space, as is possible with IAF. Volume-conserving neural architectures were first presented in in (Deco and Brauer, 1995), as a form of nonlinear independent component analysis.

Another type of normalizing flow, introduced by (Dinh et al., 2014) (*NICE*), uses similar transformations as IAF. In contrast with IAF, this type of transformations updates only half of the latent variables $\mathbf{z}_{1:D/2}$ per step, adding a vector $f(\mathbf{z}_{D/2+1:D})$ which is a neural network based function of the remaining latent variables $\mathbf{z}_{D/2+1:D}$. Such large blocks have the advantage of computationally cheap inverse transformation, and the disadvantage of typically requiring longer chains. In experiments, (Rezende and Mohamed, 2015) found that this type of transformation is generally less powerful than other types of normalizing flow, in experiments with a low-dimensional latent space. Concurrently to our work, NICE was extended to high-dimensional spaces in (Dinh et al., 2016) (*Real NVP*). An empirical comparison would be an interesting subject of future research.

A potentially powerful transformation is the *Hamiltonian flow* used in Hamiltonian Variational Inference (Salimans et al., 2014). Here, a transformation is generated by simulating the flow of a Hamiltonian system consisting of the latent variables $\mathbf{z}$, and a set of auxiliary momentum variables. This type of transformation has the additional benefit that it is guided by the exact posterior distribution, and that it leaves this distribution invariant for small step sizes. Such as transformation could thus take us arbitrarily close to the exact posterior distribution if we can apply it for a sufficient number of times. In practice, however, Hamiltonian Variational Inference is very demanding computationally. Also, it requires an auxiliary variational bound to account for the auxiliary variables, which can impede progress if the bound is not sufficiently tight.

An alternative method for increasing the flexiblity of the variational inference, is the introduction of auxiliary latent variables (Salimans et al., 2014; Ranganath et al., 2015; Tran et al., 2015) and corresponding auxiliary inference models. Latent variable models with multiple layers of stochastic variables, such as the one used in our experiments, are often equivalent to such auxiliary-variable methods. We combine deep latent variable models with IAF in our experiments, benefiting from both techniques.

# 6 Experiments

We empirically evaluate IAF by applying the idea to improve variational autoencoders. Please see appendix C for details on the architectures of the generative model and inference models. Code for reproducing key empirical results is available online[3].

## 6.1 MNIST

In this expermiment we follow a similar implementation of the convolutional VAE as in (Salimans et al., 2014) with ResNet (He et al., 2015) blocks. A single layer of Gaussian stochastic units of dimension 32 is used. To investigate how the expressiveness of approximate posterior affects performance, we report results of different IAF posteriors with varying degrees of expressiveness. We use a 2-layer MADE (Germain et al., 2015) to implement one IAF transformation, and we stack multiple IAF transformations with ordering reversed between every other transformation.

**Results:** Table 1 shows results on MNIST for these types of posteriors. Results indicate that as approximate posterior becomes more expressive, generative modeling performance becomes better. Also worth noting is that an expressive approximate posterior also tightens variational lower bounds as expected, making the gap between variational lower bounds and marginal likelihoods smaller. By making IAF deep and wide enough, we can achieve best published log-likelihood on dynamically

Table 1: Generative modeling results on the dynamically sampled binarized MNIST version used in previous publications (Burda et al., 2015). Shown are averages; the number between brackets are standard deviations across 5 optimization runs. The right column shows an importance sampled estimate of the marginal likelihood for each model with 128 samples. Best previous results are reproduced in the first segment: [1]: (Salimans et al., 2014) [2]: (Burda et al., 2015) [3]: (Kaae Sønderby et al., 2016) [4]: (Tran et al., 2015)

| Model | VLB | $\log p(\mathbf{x}) \approx$ |
|---|---|---|
| Convolutional VAE + HVI [1] | -83.49 | -81.94 |
| DLGM 2hl + IWAE [2] | | -82.90 |
| LVAE [3] | | -81.74 |
| DRAW + VGP [4] | -79.88 | |
| Diagonal covariance | -84.08 ($\pm$ 0.10) | -81.08 ($\pm$ 0.08) |
| IAF ($\mathrm{Depth} = 2, \mathrm{Width} = 320$) | -82.02 ($\pm$ 0.08) | -79.77 ($\pm$ 0.06) |
| IAF ($\mathrm{Depth} = 2, \mathrm{Width} = 1920$) | -81.17 ($\pm$ 0.08) | -79.30 ($\pm$ 0.08) |
| IAF ($\mathrm{Depth} = 4, \mathrm{Width} = 1920$) | -80.93 ($\pm$ 0.09) | -79.17 ($\pm$ 0.08) |
| IAF ($\mathrm{Depth} = 8, \mathrm{Width} = 1920$) | -80.80 ($\pm$ 0.07) | **-79.10** ($\pm$ 0.07) |

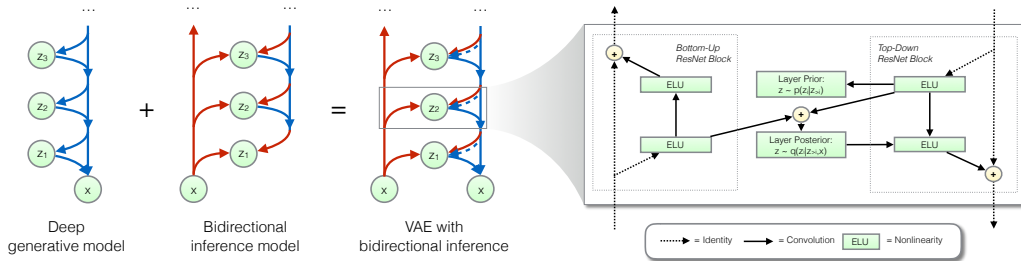

Figure 3: Overview of our ResNet VAE with bidirectional inference. The posterior of each layer is parameterized by its own IAF.

binarized MNIST: **-79.10**. On Hugo Larochelle's statically binarized MNIST, our VAE with deep IAF achieves a log-likelihood of **-79.88**, which is slightly worse than the best reported result, **-79.2**, using the PixelCNN (van den Oord et al., 2016b).

## 6.2   CIFAR-10

We also evaluated IAF on the CIFAR-10 dataset of natural images. Natural images contain a much greater variety of patterns and structure than MNIST images; in order to capture this structure well, we experiment with a novel architecture, ResNet VAE, with many layers of stochastic variables, and based on residual convolutional networks (ResNets)  (He et al., 2015, 2016). Please see our appendix for details.

**Log-likelihood.**   See table 2 for a comparison to previously reported results. Our architecture with IAF achieves **3.11 bits per dimension**, which is better than other published latent-variable models, and almost on par with the best reported result using the PixelCNN. See the appendix for more experimental results. We suspect that the results can be further improved with more steps of flow, which we leave to future work.

**Synthesis speed.**   Sampling took about **0.05 seconds/image** with the ResNet VAE model, versus **52.0 seconds/image** with the PixelCNN model, on a NVIDIA Titan X GPU. We sampled from the PixelCNN naïvely by sequentially generating a pixel at a time, using the full generative model at each iteration. With custom code that only evaluates the relevant part of the network, PixelCNN sampling could be sped up significantly; however the speedup will be limited on parallel hardware due to the

Table 2: Our results with ResNet VAEs on CIFAR-10 images, compared to earlier results, in *average number of bits per data dimension* on the test set. The number for convolutional DRAW is an upper bound, while the ResNet VAE log-likelihood was estimated using importance sampling.

| Method | bits/dim $\leq$ |
|---|---|
| *Results with tractable likelihood models*: | |
| Uniform distribution (van den Oord et al., 2016b) | 8.00 |
| Multivariate Gaussian (van den Oord et al., 2016b) | 4.70 |
| NICE (Dinh et al., 2014) | 4.48 |
| Deep GMMs (van den Oord and Schrauwen, 2014) | 4.00 |
| Real NVP (Dinh et al., 2016) | 3.49 |
| PixelRNN (van den Oord et al., 2016b) | **3.00** |
| Gated PixelCNN (van den Oord et al., 2016c) | **3.03** |
| | |
| *Results with variationally trained latent-variable models*: | |
| Deep Diffusion (Sohl-Dickstein et al., 2015) | 5.40 |
| Convolutional DRAW (Gregor et al., 2016) | 3.58 |
| ResNet VAE with IAF (Ours) | **3.11** |

sequential nature of the sampling operation. Efficient sampling from the ResNet VAE is a parallel computation that does not require custom code.

# 7   Conclusion

We presented *inverse autoregressive flow* (IAF), a new type of normalizing flow that scales well to high-dimensional latent space. In experiments we demonstrated that autoregressive flow leads to significant performance gains compared to similar models with factorized Gaussian approximate posteriors, and we report close to state-of-the-art log-likelihood results on CIFAR-10, for a model that allows much faster sampling.

# Acknowledgements

We thank Jascha Sohl-Dickstein, Karol Gregor, and many others at Google Deepmind for interesting discussions. We thank Harri Valpola for referring us to Gustavo Deco's relevant pioneering work on a form of inverse autoregressive flow applied to nonlinear independent component analysis.

## Footnotes

[2]where $\mathbf{x}$ is the context, such as the value of the datapoint. In case of models with multiple levels of latent variables, the context also includes the value of the previously sampled latent variables.

[3]`https://github.com/openai/iaf`

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
