[Supplementary Material]

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

| IAF (Depth = 2, Width = 1920) | -81.17 ($\pm$ 0.08) | -79.30 ($\pm$ 0.08) |
| IAF (Depth = 4, Width = 1920) | -80.93 ($\pm$ 0.09) | -79.17 ($\pm$ 0.08) |
| IAF (Depth = 8, Width = 1920) | -80.80 ($\pm$ 0.07) | **-79.10** ($\pm$ 0.07) |

Figure 3: Overview of our ResNet VAE with bidirectional inference. The posterior of each layer is parameterized by its own IAF.

binarized MNIST: **-79.10**. On Hugo Larochelle's statically binarized MNIST, our VAE with deep IAF achieves a log-likelihood of **-79.88**, which is slightly worse than the best reported result, **-79.2**, using the PixelCNN (van den Oord et al., 2016b).

## 6.2 CIFAR-10

We also evaluated IAF on the CIFAR-10 dataset of natural images. Natural images contain a much greater variety of patterns and structure than MNIST images; in order to capture this structure well, we experiment with a novel architecture, ResNet VAE, with many layers of stochastic variables, and based on residual convolutional networks (ResNets) (He et al., 2015, 2016). Please see our appendix for details.

**Log-likelihood.** See table 2 for a comparison to previously reported results. Our architecture with IAF achieves **3.11 bits per dimension**, which is better than other published latent-variable models, and almost on par with the best reported result using the PixelCNN. See the appendix for more experimental results. We suspect that the results can be further improved with more steps of flow, which we leave to future work.

**Synthesis speed.** Sampling took about **0.05 seconds/image** with the ResNet VAE model, versus **52.0 seconds/image** with the PixelCNN model, on a NVIDIA Titan X GPU. We sampled from the PixelCNN naïvely by sequentially generating a pixel at a time, using the full generative model at each iteration. With custom code that only evaluates the relevant part of the network, PixelCNN sampling could be sped up significantly; however the speedup will be limited on parallel hardware due to the

Table 2: Our results with ResNet VAEs on CIFAR-10 images, compared to earlier results, in *average number of bits per data dimension* on the test set. The number for convolutional DRAW is an upper bound, while the ResNet VAE log-likelihood was estimated using importance sampling.

| Method | bits/dim $\leq$ |
|---|---|
| *Results with tractable likelihood models*: | |
| Uniform distribution (van den Oord et al., 2016b) | 8.00 |
| Multivariate Gaussian (van den Oord et al., 2016b) | 4.70 |
| NICE (Dinh et al., 2014) | 4.48 |
| Deep GMMs (van den Oord and Schrauwen, 2014) | 4.00 |
| Real NVP (Dinh et al., 2016) | 3.49 |
| PixelRNN  (van den Oord et al., 2016b) | **3.00** |
| Gated PixelCNN (van den Oord et al., 2016c) | **3.03** |
| | |
| *Results with variationally trained latent-variable models*: | |
| Deep Diffusion (Sohl-Dickstein et al., 2015) | 5.40 |
| Convolutional DRAW (Gregor et al., 2016) | 3.58 |
| ResNet VAE with IAF (Ours) | **3.11** |

sequential nature of the sampling operation. Efficient sampling from the ResNet VAE is a parallel computation that does not require custom code.

## 7   Conclusion

We presented *inverse autoregressive flow* (IAF), a new type of normalizing flow that scales well to high-dimensional latent space. In experiments we demonstrated that autoregressive flow leads to significant performance gains compared to similar models with factorized Gaussian approximate posteriors, and we report close to state-of-the-art log-likelihood results on CIFAR-10, for a model that allows much faster sampling.

## Acknowledgements

We thank Jascha Sohl-Dickstein, Karol Gregor, and many others at Google Deepmind for interesting discussions. We thank Harri Valpola for referring us to Gustavo Deco's relevant pioneering work on a form of inverse autoregressive flow applied to nonlinear independent component analysis.

## Footnotes

[2]where $\mathbf{x}$ is the context, such as the value of the datapoint. In case of models with multiple levels of latent variables, the context also includes the value of the previously sampled latent variables.

[3]https://github.com/openai/iaf

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

# A   Linear IAF

Perhaps the simplest special case of IAF is the transformation of a Gaussian variable with diagonal covariance to one with linear dependencies.

Any full-covariance multivariate Gaussian distribution with mean $\mathbf{m}$ and covariance matrix $\mathbf{C}$ can be expressed as an autoregressive model with

$$y_i = \mu_i(\mathbf{y}_{1:i-1}) + \sigma_i(\mathbf{y}_{1:i-1}) \cdot \epsilon_i, \text{ with}$$
$$\mu_i(\mathbf{y}_{1:i-1}) = m_i + \mathbf{C}[i, 1:i-1]\mathbf{C}[1:i-1, 1:i-1]^{-1}(\mathbf{y}_{1:i-1} - \mathbf{m}_{1:i-1}), \text{ and}$$
$$\sigma_i(\mathbf{y}_{1:i-1}) = \mathbf{C}[i,i] - \mathbf{C}[i, 1:i-1]\mathbf{C}[1:i-1, 1:i-1]^{-1}\mathbf{C}[1:i-1, i].$$

Inverting the autoregressive model then gives $\boldsymbol{\epsilon} = (\mathbf{y} - \boldsymbol{\mu}(\mathbf{y}))/\boldsymbol{\sigma}(\mathbf{y}) = \mathbf{L}(\mathbf{y} - \mathbf{m})$ with $\mathbf{L}$ the inverse Cholesky factorization of the covariance matrix $\mathbf{C}$.

By making $\mathbf{L}(\mathbf{x})$ and $\mathbf{m}(\mathbf{x})$ part of our variational encoder we can then use this inverse flow to form a posterior approximation. In experiments, we do this by starting our with a fully-factorized Gaussian approximate posterior as in e.g. (Kingma and Welling, 2013): $\mathbf{y} = \boldsymbol{\mu}(\mathbf{x}) + \boldsymbol{\sigma}(\mathbf{x}) \odot \boldsymbol{\epsilon}$ where $\boldsymbol{\epsilon} \sim \mathcal{N}(0, \mathbf{I})$, and where $\boldsymbol{\mu}(\mathbf{x})$ and $\boldsymbol{\sigma}(\mathbf{x})$ are vectors produced by our inference network. We then let the inference network produce an extra output $\mathbf{L}(\mathbf{x})$, the lower triangular inverse Cholesky matrix, which we then use to update the approximation. With this setup, the problem is overparameterized, so we define the mean vector $\mathbf{m}$ above to be the zero vector, and we restrict $\mathbf{L}$ to have ones on the diagonal. One step of linear IAF then turns the fully-factorized distribution of $\mathbf{y}$ into an arbitrary multivariate Gaussian distribution: $\mathbf{z} = \mathbf{L}(\mathbf{x}) \cdot \mathbf{y}$. This results in a simple and computationally efficient posterior approximation, with scalar density function given by $q(\mathbf{z}|\mathbf{x}) = q(\mathbf{y}|\mathbf{x})$. By optimizing the variational lower bound we then fit this conditional multivariate Gaussian approximation to the true posterior distribution.

# B   MNIST

In MNIST expermiment we follow a similar implementation of the convolutional VAE as in (Salimans et al., 2014) with ResNet (He et al., 2015) blocks. A single layer of Gaussian stochastic units of dimension 32 is used. The inference network has three 2-strided resnet blocks with 3x3 filters and [16,32,32] feature maps. Between every other strided convolution, there is another resnet block with stride 1 and same number feature maps. There is one more fully-connected layer after convolutional layers with 450 hidden units. The generation network has a symmetric structure with strided convolution replaced by transposed convolution(Zeiler et al., 2010). When there is a change in dimensionality, we use strided convolution or transposed convolution to replace the identity connection in ResNet blocks. Exponential Linear Units (Clevert et al., 2015) are used as activation functions. The whole network is parametrized according to Weight Normalization(Salimans and Kingma, 2016) and data-dependent initialization is used.

# C   ResNet VAE

For details of the ResNet VAE architecture used for CIFAR-10, please see figure 3 and our code. The main benefits of this architecture is that it forms a flexible autoregressive prior latent space, while still being straightforward to sample from.

For CIFAR-10, we used a novel neural variational autoencoder (VAE) architecture with ResNet (He et al., 2015, 2016) units and multiple stochastic layers. Our architecture consists of $L$ stacked blocks, where each block $(l = 1..L)$ is a combination of a bottom-up residual unit for inference, producing a series of bottom-up activations $\mathbf{h}_l^{(q)}$, and a top-down residual unit used for both inference and generation, producing a series of top-down activations $\mathbf{h}_l^{(p)}$.

The hidden layer of each residual function in the generative model contains a combination of the usual deterministic hidden units and a relatively small number of stochastic hidden units with a heteroscedastic diagonal Gaussian distribution $p(\mathbf{z}_l|\mathbf{h}_l^{(p)})$ given the unit's input $\mathbf{h}_l^{(p)}$, followed by a nonlinearity. We utilize wide (Zagoruyko and Komodakis, 2016) *pre-activation residual units* (He et al., 2015) with single-hidden-layer residual functions.

Figure 4: Generative ResNet and detail of layer. This is the generative component of our ResNet VAE.

See figure 5 for an illustration of the generative Resnet. Assuming $L$ layers of latent variables, the generative model's density function is factorized as $p(\mathbf{x}, \mathbf{z}_1, \mathbf{z}_2, \mathbf{z}_3, ...) = p(\mathbf{x}, \mathbf{z}_{1:L}) = p(\mathbf{x}|\mathbf{z}_{1:L})p(\mathbf{z}_{1:L})$. The second part of this density, the prior over the latent variable, is autoregressive: $p(\mathbf{z}_{1:L}) = p(\mathbf{z}_L)\prod_{l=1}^{L-1} p(\mathbf{z}_l|\mathbf{z}_{l+1:L})$. This autoregressive nature of the prior increases the flexibility of the true posterior, leading to improved empirical results. This improved performance is easily explained: as the true posterior is largely a function of this prior, a flexible prior improves the flexibility of the true posterior, making it easier for the VAE to match the approximate and true posteriors, leading to a tighter bound, without sacrificing the flexibility of the generative model itself.

## C.1   Bottom-Up versus Bidirectional inference

Figure 5 illustrates the difference between a *bidirectional* inference network, whose topological ordering over the latent variables equals that of the generative model, and a *bottom-up* inference network, whose topological ordering is reversed.

The top left shows a generative model with three levels of latent variables, with topological ordering $\mathbf{z}_1 \rightarrow \mathbf{z}_2 \rightarrow \mathbf{z}_3 \rightarrow \mathbf{x}$, and corresponding joint distribution that factorizes as $p(\mathbf{x}, \mathbf{z}_1, \mathbf{z}_2, \mathbf{z}_3) = p(\mathbf{x}|\mathbf{z}_1, \mathbf{z}_2, \mathbf{z}_3)p(\mathbf{z}_1|\mathbf{z}_2, \mathbf{z}_3)p(\mathbf{z}_2|\mathbf{z}_3)p(\mathbf{z}_3)$. The top middle shows a corresponding inference model with reversed topological ordering, and corresponding factorization $q(\mathbf{z}_1, \mathbf{z}_2, \mathbf{z}_3|\mathbf{x}) = q(\mathbf{z}_1|\mathbf{x})q(\mathbf{z}_2|\mathbf{z}_1, \mathbf{x})q(\mathbf{z}_3|\mathbf{z}_2, \mathbf{z}_1, \mathbf{x})$. We call this a bottom-up inference model. Right: the resulting variational autoencoder (VAE). In the VAE, the log-densities of the inference model and generative model, $\log p(\mathbf{x}, \mathbf{z}_1, \mathbf{z}_2, \mathbf{z}_3) - \log q(\mathbf{z}_1, \mathbf{z}_2, \mathbf{z}_3|\mathbf{x})$ respectively, are evaluated under a sample from the inference model, to produce an estimate of the variational bound. Computation of the density of the samples $(\mathbf{x}, \mathbf{z}_1, \mathbf{z}_2, \mathbf{z}_3)$ under the generative model, requires computation of the conditional distributions of the generative model, which requires bidirectional computation. Hence, evaluation of the model requires both bottom-up inference (for sampling $\mathbf{z}$ s, and evaluating the posterior density) and top-down generation.

In case of bidirectional inference (see also (Salimans, 2016; Kaae Sønderby et al., 2016)), we first perform a fully deterministic bottom-up pass, before sampling from the posterior in top-down order in the topological ordering of the generative models.

(a) Schematic overview of a ResNet VAE with bottom-up inference.

(b) Schematic overview of a ResNet VAE with bidirectional inference.

Figure 5: Schematic overview of topological orderings of bottom-up **(a)** versus bidirectional **(b)** inference networks, and their corresponding variational autoencoders (VAEs). See section C.1

## C.2 Inference with stochastic ResNet

Like our generative Resnet, the both our bottom-up and bidirectional inference models are implemented through ResNet blocks. See figure 6. As we explained, in case of bottom-up inference, latent variables are sampled in bottom-up order, i.e. the reverse of the topological ordering of the generative model. The residual functions in the bottom-up inference network compute a conditional approximate posterior, $q(\mathbf{z}_l|\mathbf{h}_{l+1}^{(p)})$, conditioned on the bottom-up residual unit's input. The sample from this distribution is then, after application of the nonlinearity, treated as part of the hidden layer of the bottom-up residual function and thus used upstream.

In the bidirectional inference case, the approximate posterior for each layer is conditioned on both the bottom-up input and top-down input: $q(\mathbf{z}_l|\mathbf{h}_l^{(q)}, \mathbf{h}_l^{(p)})$. The sample from this distribution is, again after application of the nonlinearity, treated as part of the hidden layer of the top-down residual function and thus used downstream.

## C.3 Approximate posterior

The approximate posteriors $q(\mathbf{z}_l|.)$ are defined either through a diagonal Gaussian, or through an IAF posterior. In case of IAF, the context $\mathbf{c}$ is provided by either $\mathbf{h}_l^{(q)}$, or $\mathbf{h}_l^{(q)}$ and $\mathbf{h}_l^{(p)}$, dependent on inference direction.

We use either diagonal Gaussian posteriors, or IAF with a single step of masked-based Pixel-CNN (van den Oord et al., 2016b) with zero, one or two layers of hidden layers with ELU non-linearities (Clevert et al., 2015). Note that IAF with zero hidden layers corresponds to a linear transformation; i.e. a Gaussian with off-diagonal covariance.

We investigate the importance of a full IAF transformation with learned (dynamic) $\boldsymbol{\sigma}_t(.)$ rescaling term term (in eq.(10)), versus a fixed $\boldsymbol{\sigma}_t(.) = 1$ term. See table 3 for a comparison of resulting test-set bpp (bits per pixel) performance, on the CIFAR-10 dataset. We found the difference in performance to be almost negligible.

(a) Computational flow schematic of ResNet VAE with bottom-up inference

(b) Computational flow schematic of ResNet VAE with bidirectional inference

Figure 6: Detail of a single layer of the ResNet VAE, with the bottom-up inference (top) and bidirectional inference (bottom).

## C.4   Bottom layer

The first layer of the encoder is a convolutional layer with $2 \times 2$ spatial subsampling; the last layer of the decoder has a matching convolutional layer with $2 \times 2$ spatial upsampling. The ResNet layers could perform further up- and downsampling, but we found that this did not improve empirical results.

Table 3: CIFAR-10 test-set bpp (bits per pixel), when training IAF with location-only perturbation versus full (location+scale) perturbation.

| ResNet depth | Inference direction | Posterior | Location-only | Location+scale |
|---|---|---|---|---|
| 4 | Bottom-up | IAF with 0 hidden layers | 3.67 | 3.68 |
| 4 | Bottom-up | IAF with 1 hidden layers | 3.61 | 3.61 |
| 4 | Bidirectional | IAF with 0 hidden layers | 3.66 | 3.67 |
| 4 | Bidirectional | IAF with 1 hidden layers | 3.58 | 3.56 |
| 8 | Bottom-up | IAF with 0 hidden layers | 3.54 | 3.55 |
| 8 | Bottom-up | IAF with 1 hidden layers | 3.48 | 3.49 |
| 8 | Bidirectional | IAF with 0 hidden layers | 3.51 | 3.52 |
| 8 | Bidirectional | IAF with 1 hidden layers | 3.45 | 3.42 |

Figure 7: Shown are stack plots of the number of nats required to encode the CIFAR-10 set images, per stochastic layer of the 24-layer network, as a function of the number of training epochs, for different choices of minimum information constraint $\lambda$ (see section C.8). Enabling the constraint ($\lambda > 0$) results in avoidance of undesirable stable equilibria, and fuller use of the stochastic layers by the model. The bottom-most (white) area corresponds to the bottom-most (reconstruction) layer, the second area from the bottom denotes the first stochastic layer, the third area denotes the second stochastic layer, etc.

## C.5  Discretized Logistic Likelihood

The first layer of the encoder, and the last layer of the decoder, consist of convolutions that project from/to input space. The pixel data is scaled to the range $[0, 1]$, and the data likelihood of pixel values in the generative model is the probability mass of the pixel value under the logistic distribution. Noting that the CDF of the standard logistic distribution is simply the sigmoid function, we simply compute the probability mass per input pixel using $p(x_i|\mu_i, s_i) = \text{CDF}(x_i + \frac{1}{256}|\mu_i, s_i) - \text{CDF}(x_i|\mu_i, s_i)$, where the locations $\mu_i$ are output of the decoder, and the log-scales $\log s_i$ are learned scalar parameter per input channel.

## C.6  Weight initialisation and normalization

We also found that the noise introduced by batch normalization hurts performance; instead we use weight normalization (Salimans and Kingma, 2016) method. We initialized the parameters using the data-dependent technique described in (Salimans and Kingma, 2016).

## C.7  Nonlinearity

We compared ReLU, softplus, and ELU (Clevert et al., 2015) nonlinearities; we found that ELU resulted in significantly better empirical results, and used the ELU nonlinearity for all reported experiments and both the inference model and the generative model.

## C.8  Objective with Free Bits

To accelerate optimization and reach better optima, we optimize the bound using a slightly modified objective with *free bits*: a constraint on the minimum amount of information per group of latent

Table 4: Our results in *average number of bits per data dimension* on the test set with ResNet VAEs, for various choices of posterior ResNet depth, and IAF depth.

| Posterior: | ResNet Depth: | 4 | 8 | 12 |
|---|---|---|---|---|
| Bottom-up, factorized Gaussians | | 3.71 | 3.55 | 3.44 |
| Bottom-up, IAF, linear, 1 step) | | 3.68 | 3.55 | 3.41 |
| Bottom-up, IAF, 1 hidden layer, 1 step) | | 3.61 | 3.49 | 3.38 |
| Bidirectional, factorized Gaussians | | 3.74 | 3.60 | 3.46 |
| Bidirectional, IAF, linear, 1 step) | | 3.67 | 3.52 | 3.40 |
| Bidirectional, IAF, 1 hidden layer, 1 step) | | 3.56 | 3.42 | 3.28 |
| Bidirectional, IAF, 2 hidden layers, 1 steps) | | 3.54 | 3.39 | 3.27 |
| Bidirectional, IAF, 1 hidden layer, 2 steps) | | 3.53 | 3.36 | 3.26 |

variables. Consistent with findings in (Bowman et al., 2015) and (Kaae Sønderby et al., 2016), we found that stochastic optimization with the unmodified lower bound objective often gets stuck in an undesirable stable equilibrium. At the start of training, the likelihood term $\log p(\mathbf{x}|\mathbf{z})$ is relatively weak, such that an initially attractive state is where $q(\mathbf{z}|\mathbf{x}) \approx p(\mathbf{z})$. In this state, encoder gradients have a relatively low signal-to-noise ratio, resulting in a stable equilibrium from which it is difficult to escape. The solution proposed in (Bowman et al., 2015) and (Kaae Sønderby et al., 2016) is to use an optimization schedule where the weight of the latent cost $D_{KL}(q(\mathbf{z}|\mathbf{x})||p(\mathbf{z}))$ is slowly annealed from 0 to 1 over many epochs.

We propose a different solution that does not depend on an annealing schedule, but uses a modified objective function that is constant throughout training instead. We divide the latent dimensions into the $K$ groups/subsets within which parameters are shared (e.g. the latent feature maps, or individual dimensions if no parameter are shared across dimensions). We then use the following objective, which ensures that using less than $\lambda$ nats of information per subset $j$ (on average per minibatch $\mathcal{M}$) is not advantageous:

$$\widetilde{\mathcal{L}}_\lambda = \mathbb{E}_{\mathbf{x} \sim \mathcal{M}} \left[ \mathbb{E}_{q(\mathbf{z}|\mathbf{x})} \left[ \log p(\mathbf{x}|\mathbf{z}) \right] \right] - \sum_{j=1}^{K} \text{maximum}(\lambda, \mathbb{E}_{\mathbf{x} \sim \mathcal{M}} \left[ D_{KL}(q(\mathbf{z}_j|\mathbf{x})||p(\mathbf{z}_j)) \right]) \quad (15)$$

Since increasing the latent information is generally advantageous for the first (unaffected) term of the objective (often called the *negative reconstruction errror*), this results in $\mathbb{E}_{\mathbf{x} \sim \mathcal{M}} \left[ D_{KL}(q(\mathbf{z}_j|\mathbf{x})||p(\mathbf{z}_j)) \right] \geq \lambda$ for all $j$, in practice.

We experimented with $\lambda \in [0, 0.125, 0.25, 0.5, 1, 2, 4, 8]$ and found that values in the range $\lambda \in [0.125, 0.25, 0.5, 1, 2]$ resulted in more than $0.1$ nats improvement in bits/pixel on the CIFAR-10 benchmark.

### C.9 IAF architectural comparison

We performed an empirical evaluation on CIFAR-10, comparing various choices of ResNet VAE combined with various approximate posteriors and model depths, and inference directions. See table 4 for results. Our best CIFAR-10 result, 3.11 bits/dim in table 2 was produced by a ResNet VAE with depth 20, bidirectional inference, nonlinear IAF with 2 hidden layers and 1 step. Please see our code for further details.

## D Equivalence with Autoregressive Priors

Earlier work on improving variational auto-encoders has often focused on improving the prior $p(\mathbf{z})$ of the latent variables in our generative model. For example, (Gregor et al., 2013) use a variational auto-encoder where both the prior and inference network have recursion. It is therefore worth noting that our method of improving the fully-factorized posterior approximation with inverse autoregressive flow, in combination with a factorized prior $p(\mathbf{z})$, is equivalent to estimating a model where the prior $p(\mathbf{z})$ is autoregressive and our posterior approximation is factorized. This result follows directly from the analysis of section 3: we can consider the latent variables $\mathbf{y}$ to be our target for inference, in

which case our prior is autoregressive. Equivalently, we can consider the whitened representation $\mathbf{z}$ to be the variables of interest, in which case our prior is fully-factorized and our posterior approximation is formed through the inverse autoregressive flow that whitens the data (equation 7).