[Reviews · NeurIPS 2016]

Reviewer 1

Summary

The paper proposes a method for enriching the family of tractable variational posteriors for variational autoencoders (VAEs) by introducing a novel and efficient autoregressive density transformation (IAF) that falls under the normalizing flow framework. The transformation is efficiently parallelizable and is more expressive than the simple transformations that have been used as normalizing flows so far. The authors also introduce a ResNet-based deep convolutional VAE architecture for modeling images and evaluate it on CIFAR-10, achieving highly competitive results using IAF.

Qualitative Assessment

This is a well written paper based on a simple but potentially powerful idea of obtaining invertible mappings with efficiently computable determinants by inverting autoregressive transformations. This allows taking advantage of the power of autoregressive transformations without paying the price of slow sequential sampling. The approach is clearly explained and is to the best of my knowledge novel. It would have been good to include a discussion of the relationship of the proposed approach to autoregressive connections between latent variables in the prior and/or variational posterior. A discussion of the computational cost of the proposed as well as the alternative approaches would have also been welcome. The paper would benefit from an empirical comparison to the previously introduced simpler normalizing flows (Eq. 6). It is not obvious that comparable results cannot be achieved by chaining many simpler (and faster) transformations together. Aside from this omission the experimental section is interesting and well done, clearly showing that IAF improves substantially on diagonal posteriors. The CIFAR results are impressive. The claim on l. 270 that their model outperforms models other than PixelRNN on CIFAR even when using a diagonal posterior is not true however, as both PixelCNN and Convolutional DRAW perform better according to Table 3.

Confidence in this Review

3-Expert (read the paper in detail, know the area, quite certain of my opinion)


Reviewer 2

Summary

In this paper the authors propose a new way to define and parametrise the approximate inference model for VAE-style models. Using a simple but clever parametrization, the proposed method allows to model non-factorial approximate posterior distributions without paying the (on modern computer architectures typically high) price of requiring sequential computation, which is otherwise associated with autoregressive models.

Qualitative Assessment

The paper is well written and the motivation behind the approach well explained. The idea of using inverse autoregressive flows is interesting and the introduction nicely connects it to various previously published approaches. In the experimental section, it is not obvious which MNIST dataset has been used: the statically sampled version by Hugo Larochelle, or the dynamically sampled one that was (re-)introduced by the Importance-weighted Autoencoders (IWAE) paper. The CIFAR-10 experiments try to disentangle the improvements from the presented method, and from the various other architectural/optimizational features of the investigated models; but according to table 2, the improvements from the inverted autoregressive flow seem rather small. Smaller than expected and small compared to the various other architectural changes. It would also be interesting to empirically compare the proposed method to full autoregressive approximate posterior distributions.

Confidence in this Review

2-Confident (read it all; understood it all reasonably well)


Reviewer 3

Summary

The authors propose an extension of normalizing flows for improving the expressivity of the approximating distribution during variational inference. They develop inverse autoregressive flows. It takes the idea of autoregressive networks in latent variable modeling and inverts it for autoregression in the data transform of variational distributions. Using whitening, the procedure can be done at once rather than sequentially.

Qualitative Assessment

The idea is interesting, particularly in its approach to enable parallel computation on GPUs by whitening the data. This seems like a practical approach to choosing the family of transformations in the normalizing flows framework. Its novelty is low based on whitening, and being a simple extension to just one of many approaches for building expressive variational families. The experiments are lacking in comparison to the slew of recent approaches to expressive approximating families for variational inference. It mostly compares to non-variational inference based approaches and traditional approaches (e.g., diagonal Gaussians). For example, how does it compare to planar or radial flows as in the original paper? And the results are not that convincing even when compared to only these. The paper should also comment on how it relates to additional variational inference works (and possibly add these as baselines for comparison), e.g., mixtures (Jaakkola and Jordan, 1998) and variational Gaussian processes (Tran et al., 2016). Inverse autoregressive flows also sounds broadly useful in the scope of auxiliary variable techniques, e.g., hierarchical variational models (Ranganath et al., 2016) and auxiliary deep generative models (Maaloe et al., 2016). This could be a useful application broadening tis scope beyond deep generative models with differentiable latent variables, especially those with only Gaussian latent variables

Confidence in this Review

3-Expert (read the paper in detail, know the area, quite certain of my opinion)


Reviewer 4

Summary

This paper proposes "inverse autoregressive flow", a technique that transforms a simple distribution into a more flexible posterior distribution for variational inference. IAF can be viewed as a simplified form of the normalizing flow (Rezende and Mohamed ,2015), where autoregressive functions are used as nonlinear transformations to make the posterior approximation for variational inference more flexible. The transformation has simple Jacobian determinant so it’s computationally efficient. In this paper, three variants of the IAF have been introduced and experimented: linear IAF, nonlinear IAF through MADE and nonlinear IAF through recurrent neural networks. The model is validated on MNIST and CIFAR-10 database. The proposed VAE+IAF outperforms vanilla VAE with simple diagonal Gaussian distribution in terms of log-likelihood. The samples generated from the VAE trained with IAF prior look less blurry compared to previous work using VAE.

Qualitative Assessment

This paper proposes a general solution to the variational auto-encoders (VAE) that enables more complicated and structured prior distribution. The experimental comparisons with previous work are solid enough. The paper is in general clearly written but the writing and formulation still needs polishing. -- Line 150 -- 151 "where at each step we use a differently parametrized auto-regressive model \mathcal{N}(\mu^t, \sigma^t)" My understanding is that only z^0 is generated by diagonal Gaussian distribution \mathcal(\mu^0, \sigma^0), while the remaining z^t is obtained by transformation step z^t = L(x) \cdot z^{t-1}. Can you clarify the pipeline for generating a sample from VAE+IAF? -- Line 158 -- Line 159: The current notation is quite confusing. The z^t has been defined in Eq.11 but not used here. It would be more clear if the author provide the final form rather than using "y" here. Also, "y" is often used for label rather than latent variable. -- How many IAF steps are used in the experiments? The only clue I find is the notation "z^l" (line 227), where "l" is the l-th block (1 = 1..L). Please elaborate on this point. -- How does the proposed IAF compared to normalization flow (Rezende and Mohamed, 2015)? The previous work (Figure 3 of that paper) clearly demonstrated that proposed technique produces better approximation to true posterior. Please comment on the performance of IAF.

Confidence in this Review

2-Confident (read it all; understood it all reasonably well)


Reviewer 5

Summary

The paper extends the work on tractable probability transformation for amortized variational inference such as Normalizing Flows. In this paper, they are proposing another class of tractable transformation: inverse autoregressive flows. Usually used for distribution estimation, inverse autoregressive flows obtain their tractability from their triangular structure. Moreover, this paper capitalize on the recent advances on autoregressive models for distribution estimation in RIDE and PixelRNN/CNN, to build powerful and computationally efficient inverse autoregressive flows. The authors obtain marginal improvement on MNIST and significant improvement on CIFAR-10.

Qualitative Assessment

The paper are able to exploit the recent advances in autoregressive models, particularly in making efficient inference through parallel computing. However, they avoid the cumbersome sampling/inversion procedure of autoregressive model, which is quite ingenious. The model description seems detailed enough, although several architectural choices are not explained (e.g. using only one IAF iteration for CIFAR-10 or using logistic distribution instead of categorical for p(x|z)). The improvement on MNIST is reasonable but not that significant, but MNIST is relatively small scale. The improvement on CIFAR-10 is quite significant. Further experiments would be necessary to fully show the potential of this method on image dataset. Imagenet would be an example of a dataset to try on.

Confidence in this Review

3-Expert (read the paper in detail, know the area, quite certain of my opinion)


Reviewer 6

Summary

This paper proposes an extension to the variational autoencoder (VAE) framework by making the posterior more expressive. A simple posterior from vanilla VAEs is transformed into a more complicated posterior distribution by successively applying an invertible autoregressive function. The log density of such transformation functions can be efficiently computed as long as the Jacobian determinant of each such transformations can be computed. Authors propose a specific form of such a function which allows computation of the Jacob-det with linear scaling w.r.t dimensions. This method is directly related to the normalizing flows work by Rezende & Mohamed, which is a general theoretical framework for iteratively constructing expressive posteriors. IAF is a particular type of a flow function, with several appealing properties. The main contribution can be summarized as follows: IAF makes VAEs more expressive by transforming simple posteriors into more complicated ones by applying a series of invertible transformations (flow functions) within an autoregressive framework.

Qualitative Assessment

- This paper is very well written and I have little to say about its clarity and organization. - Although IAFs don’t produce state of the art log-l results on MNIST or CIFAR-10, it provides a simple and elegant method to build deep generative models with expressive posteriors. They also seem to provide a way to scale normalizing flows in the context of deep generative models. - In Table 3, how much of the performance gain is due to resnet structure vs IAF? Or inference network structure vs expressivity of the posterior? - Pixel RNNs outperform IAFs. Perhaps the authors should discuss runtime performance/time complexity of both these approaches. - In Fig 1, authors should probably show actual images from CIFAR-10 next to the samples. All deep learning people will know about it. However, I feel that the paper might have a wider appeal and it would be helpful for those people.

Confidence in this Review

2-Confident (read it all; understood it all reasonably well)